# Ibrutinib Inhibits BTK Signaling in Tumor-Infiltrated B Cells and Amplifies Antitumor Immunity by PD-1 Checkpoint Blockade for Metastatic Prostate Cancer

**DOI:** 10.3390/cancers15082356

**Published:** 2023-04-18

**Authors:** Gengguo Deng, Jiannan He, Qunxiong Huang, Tengcheng Li, Zhansen Huang, Shuntian Gao, Jinbin Xu, Tiantian Wang, Jinming Di

**Affiliations:** 1Department of Urology, The Third Affiliated Hospital, Sun Yat-sen University, Guangzhou 510630, China; 2Department of Urology, The Sixth Affiliated Hospital, Sun Yat-sen University, Guangzhou 510655, China; 3Department of Infertility and Sexual Medicine, The Third Affiliated Hospital, Sun Yat-sen University, Guangzhou 510630, China; 4Department of Medical Oncology, The Third Affiliated Hospital, Sun Yat-sen University, Guangzhou 510630, China

**Keywords:** ibrutinib, BTK, prostate cancer, PD-1, B cells, CD8^+^ T cells

## Abstract

**Simple Summary:**

Metastatic prostate cancer has a poor clinical prognosis and is incurable. In recent decades, immune checkpoint blockade has revolutionized anticancer therapies and represents a potential way to control metastatic prostate cancer. Unfortunately, due to low immunoreactivity, patients failed to achieve clinical benefits from PD-1/PD-L1 blockade. The underpinning mechanisms involved in the drug resistance of immune checkpoint blockade focus the research for metastatic prostate cancer treatment. In this study, we provide a novel combinational regimen with a kinase inhibitor and PD-1/PD-L1 axis blockade to overcome the above-mentioned drug resistance. We believe that our research will be helpful for the development of novel strategies for metastatic or late-stage prostate cancer immunotherapy.

**Abstract:**

Metastatic prostate cancer (PCa) remains incurable and causes considerably diminished overall survival. Despite significant progress in pharmacotherapy, the disease prognosis remains unchanged. Immune checkpoint inhibitors (ICIs) have demonstrated effectiveness in treating various advanced malignancies, but their efficacy in metastatic PCa is relatively limited. Previous studies have confirmed the immunosuppressive role of tumor-infiltrating B cells (TIL-Bs) in the PCa microenvironment, which accounts for their poor immunogenic potency. In this study, we demonstrated that an oral kinase agent, ibrutinib, strongly potentiated anti-PD-1 checkpoint blockade efficacy and successfully controlled tumor growth in a murine orthotopic PCa model constructed using a metastatic and hormone-independent cell line (RM-1). We identified close relationships between TIL-Bs, Bruton’s tyrosine kinase (BTK), and immunosuppressive molecules by bioinformatics and histological analysis. An in vitro study showed that a low dose of ibrutinib significantly inhibited B cell proliferation and activation as well as IL-10 production through the BTK pathway. Moreover, ibrutinib-treated B cells promoted CD8^+^ T cell proliferation and inhibitory receptor (IR) expression. However, the same dose of ibrutinib was insufficient to induce apoptosis in cancer cells. An in vivo study showed that ibrutinib monotherapy failed to achieve tumor regression in murine models but decreased B cell infiltration and inhibited activation and IL-10 production. More importantly, CD8^+^ T cell infiltration increased with high IR expression. Ibrutinib synergized with anti-PD-1 checkpoint blockade enormously improved antitumor immunity, thereby reducing tumor volume in the same scenario. These data set the scene for the clinical development of ibrutinib as an immunogenic trigger to potentiate anti-PD-1 checkpoint blockade for metastatic PCa immunotherapy.

## 1. Introduction

Prostate cancer (PCa) is a leading cause of morbidity and mortality among men worldwide [1]. Although localized PCa has a more favorable prognosis, metastatic PCa only has a 30% 5-year cancer-specific survival rate and is considered incurable [2]. For metastatic PCa therapy, androgen deprivation therapy and chemotherapy have long been standard treatments. However, a majority of patients will fail treatment and eventually progress to castration-resistant prostate cancer (CRPC). Although significant advances have been made in the field’s biological understanding and the development of new drugs for the treatment of metastatic PCa in the past two decades, no curative approach has been established [3,4,5]. Most men with metastatic PCa will finally die from their disease. Hence, further exploration is needed to identify novel and efficient therapeutic approaches for metastatic PCa.

In recent decades, immunotherapy has become a promising approach for treating many types of cancer and has changed the treatment landscape [6]. Over the past few decades, research on cancer immunotherapy using immune checkpoint inhibitors (ICIs) targeting the programmed cell death-1 (PD-1)/programmed cell death-ligand 1 (PD-L1) pathway has achieved remarkable progress. For PCa, immunotherapy alone has shown limited clinical benefits for metastatic CRPC (mCRPC). A clinical trial showed that patients with mCRPC who received either anti-PD-1 inhibitor monotherapy or anticytotoxic T-lymphocyte-associated protein 4 (CTLA4) inhibitor monotherapy did not demonstrate significant response rates. Combination therapy with an anti-PD-1 inhibitor and an anti-CTLA4 inhibitor only showed modest responses in a small number of mCRPC patients [7,8,9]. PCa has long been considered the “cold tumor” with an immunosuppressive tumor microenvironment (TME) and scarce lymphocyte infiltration. Hence, it is less sensitive to checkpoint blockade therapy [10]. Therefore, therapeutic regimens of combined ICIs with other therapeutic methods to reverse PCa from a “cold tumor” to “hot tumor” for strengthening antitumor immunity remain highly in need.

In recent years, a series of reports have highlighted the protumorigenic role of tumor-infiltrating B cells (TIL-Bs) [11,12]. Clinicians have confirmed that more intra-tumoral B cells are present in radical prostatectomy specimens from high-risk patients and those who eventually have PCa recurrence or progression [13]. Mechanically, on the one hand, androgen ablation causes B cell infiltration, and the TIL-Bs produce lymphotoxin by IKKβ activation to enhance CRPC cell survival through IKKα and State3 activation [14]. On the other hand, the recurrence of PCa after androgen ablation and androgen-induced prostate regeneration is dependent on B lymphocytes that activate IKKα within tissue progenitors. Subsequently, IKKα induces the phosphorylation and nuclear translocation of E2F1 to enhance its recruitment to the promoters/regulatory regions of the Bmi1 and Ccne genes, thereby promoting progenitor cell proliferation [15]. From the perspective of immunological mechanisms, a subset of TIL-Bs in both mouse and human PCa were identified as immunosuppressive IgA^+^ plasmocytes that modulate the response to immunogenic chemotherapy. These cells express IgA, interleukin-10 (IL-10), and PD-L1 depending on TGF-β signaling and suppress cytotoxic T cell-dependent eradication of oxaliplatin-treated PCa [16]. Accordingly, specific TIL-B subsets in the PCa microenvironment behave as tumor promoters, the elimination of which helps promote tumor regression.

Bruton’s tyrosine kinase (BTK) is a non-receptor kinase that plays a critical role in regulating B cell development and functions [17]. Ibrutinib, also known as PCI-32765, is an orally available small-molecule inhibitor that irreversibly blocks BTK activity [18]. Although the impressive efficacy of ibrutinib has mainly been shown in clinical applications for hematological malignancies or B cell malignancies, it has been extended to the field of solid tumors [19,20]. Recent findings have demonstrated strong expression of BTK in human metastatic PCa tissues. Meanwhile, ibrutinib was discovered to inhibit the proliferation, migration, and invasion of PCa cells but also decreased the synthesis of matrix metalloproteinase-2 and -9 in vitro [21]. However, in vivo evidence regarding ibrutinib treatment of PCa is limited. The therapeutic potential of combined treatment with immunotherapy for PCa remains unknown and requires further investigation.

In this study, we aimed to investigate the therapeutic efficacy in metastatic PCa of suppressing the survival and function of TIL-Bs by inhibiting BTK signaling using ibrutinib and how this affects the PCa TME to sensitize anti-PD-1 blockade. To our knowledge, this is the first study to show therapeutic efficacy of combined use of a BTK inhibitor and PD-1 blockade for metastatic PCa. Our findings suggested that TIL-Bs were immunosuppressive in the metastatic PCa TME and can be inhibited by ibrutinib to enhance PD-1 blockade immunotherapy.

## 2. Materials and Methods

### 2.1. Tissue Samples

Tumor tissues from patients who underwent radical prostatectomy and were pathologically diagnosed with localized and metastatic PCa (6 of each) from the Third Affiliated Hospital of Sun Yat-sen University were collected in 2022. These patients had not received chemotherapy or radiotherapy before radical prostatectomy. The tissue samples were fixed with 4% paraformaldehyde, paraffin-embedded, cut into paraffin sections (4 μm), and stored until use. Informed consent was obtained from all participants. The study was conducted in accordance with the Declaration of Helsinki and approved by the Human Ethics Committee of the Third Affiliated Hospital of Sun Yat-sen University.

### 2.2. Public Data Acquisition and Preprocessing

Gene correlation analysis among immune cell biomarkers was performed and evaluated using the well-known online database TIMER (https://cistrome.shinyapps.io/timer/, accessed on 25 January 2022) [22]. TIMER allows us to systematically assess relationships among immune cell-related genes in different types of human cancer. We used TIMER to analyze the correlation among CD79A, BTK, PDCD1, CTLA4, HAVCR2, and IL-10 in PCa. Single-cell RNA sequencing data were obtained from an online scRNA-seq database—the Tumor Immune Single Cell Hub (TISCH, http://tisch.comp-genomics.org/, accessed on 25 January 2022) [23]. The TISCH provides processed scRNA-seq data at the single-cell level to explore the TME across different human cancer types. We downloaded a dataset (GSE143791) containing metastatic prostate cancer samples and reanalyzed it with R packages in Bioconductor (http://bioconductor.org/, accessed on 25 January 2022) to explore BTK expression in B cells at the single-cell level in metastatic PCa.

### 2.3. Mice

Wild-type C57BL/6 mice (6–8 weeks old, male, and weighing 20–25 g) were purchased from the Guangdong Medical Laboratory Animal Center (Guangzhou, China) and housed under standard pathogen-free conditions in the animal facility of Sun Yat-sen University. All animal procedures were conducted with the approval of the Institutional Animal Care and Use Committee at Sun Yat-sen University.

### 2.4. Cell Lines

The human prostate cancer cell line PC-3 and the mouse prostate cancer cell line RM-1 were both purchased from the American Type Culture Collection (ATCC) and cultured in RPMI-1640 medium supplemented with 10% fetal bovine serum (FBS) and antibiotics (100 U/mL penicillin and 100 μg/mL streptomycin) in a humidified atmosphere at 37 °C and 5% CO_2_.

### 2.5. Lymphocyte Isolation and Culture

For the in vitro study, mouse spleens were mechanically disrupted, homogenized, and lysed with red blood cell lysis buffer (Solarbio Life Sciences, Beijing, China, Cat# R1010) to obtain single-cell suspensions. Bulk CD3 T cells and CD19 B cells were purified from the prepared spleen suspensions by magnetic microbead isolation kits (Miltenyi Biotec, Bologna, Italy, Cat# 5180108168 and Cat# 5200308570, respectively) and the AutoMACS program according to the manufacturer’s instructions. For the in vivo study, tumor tissues were minced into pieces of 1–2 mm and digested with collagenase IV (1.25 mg/mL, Sigma, St. Louis, MI, USA, C5138), soybean trypsin inhibitor (0.1%, Sigma, T9003), hyaluronidase (1 mg/mL, Sigma, CAS37326-33-3), and DNase I (100 mg/mL, Sigma, D5025) in DMEM at 37 °C for 40 min with shaking at 80 r/minutes. Cell suspensions were filtered over a 70 μM screen and subjected to Ficoll density gradient centrifugation to isolate tumor-infiltrating lymphocytes (TILs). Tumor samples from localized and metastatic PCa patients were minced and digested as described above. TILs were isolated with Ficoll density gradient centrifugation and purified with magnetic microbead isolation kits (Miltenyi Biotec, Cat# 130-050-301) to obtain CD19 B cells. All isolated cells were collected in the complete RPMI medium containing 10% FCS with 1% penicillin–streptomycin for further experiments.

### 2.6. Western Blot

Purified CD19 B cells from mouse spleens were stimulated with 10 μg/mL lipopolysaccharide (LPS, Sigma, L2880) for 72 h at 5% CO_2_ and 37 °C. The total protein of the activated B cells was then extracted by RIPA lysis buffer (Beyotime, Shanghai, China, P0013B) containing phosphatase inhibitor (Beyotime, P1050) and protease inhibitor (Beyotime, P1011). The prepared protein was separated on 10% SDS-PAGE gels and transferred on polyvinylidene fluoride membranes (0.45 µm; Millipore, Burlington, MA, USA, IPVH00010). Antibodies specific to Btk (Cell Signaling Technology, Danvers, MA, USA, 8547S), pBtk (Cell Signaling Technology, 87141S), PLCγ2 (Cell Signaling Technology, 34264S), pPLCγ2 (Cell Signaling Technology, 3871S), ERK (Cell Signaling Technology, 4695S), pERK (Cell Signaling Technology, 4370S), NF-κB p65 (Cell Signaling Technology, 4764S), pNF-κB p65 (Cell Signaling Technology, 3033S), and GAPDH (Cell Signaling Technology, 8884S) were incubated with the membranes overnight at 4 °C. The membranes were incubated with secondary peroxidase-conjugated antibodies the next day at room temperature for 1 h. Chemiluminescent signals were detected by the ECL method (Beyotime, Shanghai, China, P0018FS). The relative expression of each band was calculated with Fiji software.

### 2.7. Mouse Models of Orthotopic Prostate Cancer and In Vivo Treatment

RM-1 cells were cultured as described above and harvested in the logarithmic growth phase. Then, the cells were resuspended in a mixed solution composed of PBS and Matrigel (BD Biosciences, Franklin Lakes, NJ, USA, 354234) at a ratio of 3:1. Mice were anesthetized and injected with a total of 2 × 10^5^ RM-1 cells in 15 µL mixed solution into the right anterior prostate lobe using a 20 μL microsyringe. When tumors were palpable (7 days), the mice were randomized and received treatment. Ibrutinib (Selleckchem, Houstan, TX, USA, S2680) was diluted in 10% hydroxypropyl beta-cyclodextrin (Selleckchem, C12959627) and administered once a day at 6 mg/kg per mouse intraperitoneally. PD-1 blocking antibody (RMP1-14, BioXcell, Lebanon, NH, USA, BE0146) was diluted in PBS and administered intraperitoneally every three days at 150 μg/dose per mouse. Tumor volume was calculated as Volume = Length × Width^2^/2 at the indicated time points. Tumor weight was measured after sacrifice. Mice were sacrificed on day 21 after treatment began, and tumors were harvested and cut into several fragments for further experiments. Six mice from each group were randomly selected for survival analysis.

### 2.8. Histopathological Assessment

H&E staining was applied for the morphological evaluation of organs and tumors. Briefly, tumors and organs, including hearts, livers, spleens, lungs, and kidneys, were excised from the sacrificed mice. The above tissues were fixed with 4% paraformaldehyde, decalcified, dehydrated, cleared with xylenes, infiltrated with molten paraffin, and embedded in paraffin blocks. Four-micron-thick sections were cut and stained with hematoxylin and eosin. Section images were acquired and observed under a light microscope.

### 2.9. Immunohistochemistry Assessment

Immunohistochemistry (IHC) staining was performed to evaluate Ki67 expression in tumor sections in different groups. Briefly, the paraffin-embedded tumor sections were baked for 2 h at 60 °C, deparaffinized in xylene, and rehydrated in graded alcohol solutions and water. After heat-induced epitope retrieval, the sections were subjected to deactivation of endogenous peroxidase with 3% H_2_O_2_ for 10 min and non-specific blocking with the Immunol Staining Blocking Buffer (Beyotime, P0102) for 30 min at room temperature. Then, the sections were incubated with a primary antibody recognizing Ki67 (Abcam, Cambridge, UK, ab15580) at 4 °C overnight. On the next day, horseradish-peroxidase-conjugated secondary antibodies and diaminobenzidine (Gene Tech, San Francisco, CA, USA, GK500710) were added to detect the bound anti-Ki67. Hematoxylin (Keygen Biotech, Nanjing, China) was used to stain nuclei. Section images were acquired and observed under a light microscope.

### 2.10. Immunocytofluorescence and Confocal Microscopy

Cells were washed and fixed with 4% paraformaldehyde for 5 min. After washing, the fixed cells were incubated with 0.1% Triton-X100 (Sigma, T8787) and 3% BSA in PBS for 30 min at room temperature. Then, the cells were incubated with primary antibodies for CD19 (eBioscience, San Diego, CA, USA, 14-0194-80), pBTK (Bioss antibodies, Woburn, MA, USA, bs-3069R), and IL-10 (ABclone, Woburn, MA, USA, a2171) at 4 °C overnight. On the next day, the cells were washed and incubated with secondary antibodies (Invitrogen, Waltham, MA, USA, A-11006 and A-11012) at 37 °C for 1 h. The nuclei were counterstained with DAPI (Beyotime, P0131). Confocal images were photographed and investigated under a confocal microscope (LSM880, Zeiss, Oberkochen, Germany).

### 2.11. Multiplex Immunohistochemistry and Confocal Microscopy

Multiplex immunohistochemistry (mIHC) was performed using a tyramide signal amplification (TSA) fluorescence staining kit (Servicebio, Woburn, MA, USA, G1226-100T) according to the manufacturer’s instructions. Briefly, sections were subjected to heat-induced epitope retrieval, deactivation of endogenous peroxidase, and incubated with a primary antibody for CD20 (Abcam, ab78237), pBTK (Bioss antibodies, bs-3069R), CD19 (Abcam, ab245235), or CD8 (Abcam, ab237709 for human tissues, ab217344 for mouse tissues) overnight at 4 °C. Secondary antibodies (Gene Tech, GK500710) were then used on the next day, followed by fluorophore-conjugated TSA buffer: iF488 and iF555. Heat-induced epitope retrieval and antibody incubation steps were repeated for a second primary antibody and TSA. Finally, the sections were stained with DAPI for 10 min at room temperature in the dark prior to coverslipping in the antifade mountant (Beyotime, P0131). Confocal images were photographed and investigated under a confocal microscope (LSM880, Zeiss, Oberkochen, Germany).

### 2.12. Flow Cytometry

For cell surface staining, the isolated lymphocytes were washed three times with PBS buffer and resuspended in 100 μL PBS (5 × 10^5^ cells/tube). Antimouse CD16/CD32 (553141, BD Biosciences) was used for non-specific Fc block. The cell suspensions were stained with fluorochrome-conjugated antibodies, including BV421 antimouse CD45 (Biolegend, San Diego, CA, USA, 103134), APC/Cyanine7 antimouse CD3 (Biolegend, 100361), APC antimouse CD8 (Biolegend, 100712), PE antimouse PD-1 (Biolegend, 109104), BV605 antimouse CTLA4 (Biolegend, 106323), PerCP/Cyanine5.5 antimouse CD19 (Biolegend, 115534), PE/Cyanine7 antimouse CD80 (Biolegend, 104734), APC antihuman CD45 (Biolegend, 982304), FITC antihuman CD3 (Biolegend, 317305), BV421 antihuman CD8 (Biolegend, 344747), and PE antihuman PD-1 (Biolegend, 379209) in the dark for 40 min at 4 °C.

For detection of intracellular IL-10 in B cells, the cells were stimulated with 10 μg/mL LPS (Sigma, L2880), 50 ng/mL phorbol myristate acetate (PMA, Sigma, P1585), 1 μg/mL ionomycin (Sigma, I9657), and 1 μg/mL Brefeldin A (BFA, Sigma, B5936) for 5 h at 5% CO_2_ and 37 °C. For detection of intracellular interferon-γ (IFN-γ) in T cells, the cells were stimulated with 50 ng/mL PMA, 1 μg/mL ionomycin, and 1 μg/mL BFA for 5 h at 5% CO_2_ and 37 °C. After cell surface staining, the cells were fixed with fixation buffer (Biolegend, 420801), permeabilized with the permeabilization wash buffer (Biolegend, 421002) according to the manufacturer’s instructions, and then stained with APC antimouse IL-10 (Biolegend, 505010) or PE antimouse IFN-γ (Biolegend, 505808). Data were acquired on a FACSCalibur cytometer (BD Biosciences) and analyzed using CytExpert 2.4.0.28. Gating strategies are illustrated in Appendix A (in vitro study) and Appendix A (in vivo study).

### 2.13. CCK8 Assay for Cell Viability

PC-3 (2 × 10^3^/well) or RM-1 (1 × 10^3^/well) cells were seeded in a 96-well plate overnight and treated with different concentrations of ibrutinib (0, 1, and 5 μM) the next day. The culture medium of cells was removed after 24, 48, and 72 h. Then, the cells were incubated with CCK8 solution (MedChemExpress, HY-K0301) for 2 h at 37 °C in the dark. Absorbance was measured using a microplate reader (Thermo Scientific, Fremont, CA, USA) at 450 nm. The measured OD values were converted into cell viability according to the manufacturer’s protocol.

### 2.14. Apoptosis Detection

PC-3 and RM-1 cells were seeded in 24-well plates and treated with ibrutinib at 0, 1, and 5 μM for 48 h, followed by flow cytometry analysis using an Annexin V-FITC Apoptosis Detection Kit (Thermo Fisher Scientific, Cat# BMS500FI/100). In detail, cells were harvested and resuspended in the binding buffer. PI and Annexin V-FITC were then added to the resuspended cells for 15 min at room temperature. The fluorescence of the cells was read by flow cytometry in 20 min. The percentages of early and late apoptotic cells are presented in bar plots.

TUNEL staining was performed to detect apoptotic cells in tumor sections using a TUNEL kit (Beyotime Institute of Biotechnology, C1086). Briefly, paraffin sections (4 μm) were dewaxed with dimethylbenzene and rehydrated with ethanol solution (100%–75%) and distilled water. Then, the sections were incubated in 20 μg/mL proteinase K for 25 min at 37 °C. After washing with PBS three times, the sections were incubated with reaction buffer according to the manufacturer’s instructions for 60 min at 37 °C in the dark. Finally, the sections were stained with DAPI for 5 min and observed under a fluorescence microscope (Olympus Inc., Tokyo, Japan). The average TUNEL-positive cells in four random microscopic fields (100×) in each section were calculated with Fiji software. 

### 2.15. Determination of B and T Cell Proliferation

Splenic CD19^+^ B cell proliferation was determined by flow cytometry using a FITC- bromodeoxyuridine (BrdU) Flow Kit (BD Pharmingen, Franklin Lakes, NJ, USA, 559619). CD19^+^ B cells were first incubated with 10 μM BrdU at 37 °C in 5% CO_2_ for 90 minutes and then fixed and permeabilized. DNase was applied to uncouple DNA strands for 60 min at 37 °C in the dark. After washing, the cells were then stained with FITC-anti-BrdU antibody for 60 min at 4 °C in the dark. The frequency of CD19^+^BrdU^+^ B cells was measured by flow cytometry.

Splenic CD3^+^ T cell proliferation was determined by flow cytometry using the carboxyfluorescein diacetate succinimidyl ester (CFSE) method. Briefly, purified CD3^+^ T cells were washed and labeled with CFSE (Invitrogen, C34554) at a concentration of 5 μM at 37 °C in the dark for 15 min. The labeling reaction was stopped by adding five times the volume of precooled PBS. Then, the cells were washed and stimulated with anti-CD3 (BD Pharmingen, Franklin Lakes, NJ, USA, 555336) and anti-CD28 (BD Pharmingen, 555725), followed by co-culturing with ibrutinib-pretreated CD19^+^ B cells at 37 °C in 5% CO_2_ for 72 h. Upon harvest, CD3^+^ T cells were washed and stained with BV421 antimouse CD45 (Biolegend, 103134), APC/Cyanine7 antimouse CD3 (Biolegend, 100361), and APC antimouse CD8 (Biolegend, 100712) on ice for 30 min. Flow cytometry was performed to detect CD8^+^ T cell proliferation.

### 2.16. Statistical Analysis

Statistical analysis was performed by GraphPad Prism 8. Data are presented as the mean ± SD. Statistical differences between groups of mice were determined by using Student’s *t*-test. The Kaplan–Meier model and log–rank test were applied for survival analysis. A *p* value less than 0.05 was considered statistically significant (*, *p* < 0.05; **, *p* < 0.01; ***, *p* < 0.001; ****, *p* < 0.0001).

## 3. Results

### 3.1. BTK Is Positively Correlated with CD79A and Immunosuppressive Molecules and Is Largely Phosphorylated in TIL-Bs in Human Metastatic PCa

To investigate the relationship between BTK, total TIL-Bs, and immunosuppressive molecules in the PCa microenvironment, we performed gene correlation analysis using the TIMER online database. CD79A is a well-known surface marker for B cells, and its expression represents the total number of TIL-Bs in the PCa microenvironment. Gene correlation analysis revealed that CD79A expression was significantly associated with PDCD1, CTLA4, HAVCR2, and IL-10, which are commonly recognized as immunosuppressive molecules (Figure 1A). From a transcriptomic perspective, these results indicate that TIL-Bs probably contribute to the immunosuppressive environment in PCa. Consistent with the notion that BTK is critical for B cell survival, we found that BTK expression was significantly correlated with CD79A expression. Similar to CD79A, BTK expression was also strongly correlated with immunosuppressive molecules. While PDCD1, CTLA4, HAVCR2, and IL-10 are not directly related to the BTK pathway, they can potentially interact with or modulate immune responses that involve B cells and the BTK pathway. The specific mechanisms and interactions between these genes and the BTK pathway are still an area of active research. The above results emphasize that TIL-Bs probably exhibit an immunosuppressive property via the BTK pathway and affect T cell function in the prostate cancer microenvironment. As we sought to optimize the effect of immunotherapy for metastatic PCa, a single-cell RNA sequencing dataset containing metastatic PCa information was downloaded from a public database and reanalyzed to confirm BTK expression in TIL-Bs in the metastatic PCa microenvironment. The results demonstrated that BTK was largely expressed in TIL-Bs (Figure 1B). As is known, only activated kinases can transmit downstream signaling, so we examined BTK phosphorylation in PCa tissues by mIHC staining. As shown in Figure 1C, phosphorylated BTK in TIL-Bs was mainly observed in metastatic PCa tissues compared with localized ones. In addition, we isolated TILs from localized and metastatic prostate cancers to perform immunofluorescence and further prove different BTK phosphorylation and the well-known immunosuppressive molecule PD-1 expression in B cells (Appendix A). We purified CD19^+^ B cells from the isolated TILs and treated them with ibrutinib and IL-10 production in B cells was found decreased (Appendix A). These results imply that controlling BTK activity in metastatic PCa is more likely to intervene in B cell immunity to inhibit tumor maintenance. Taken together, these results indicate that BTK activity in TIL-Bs favors the induction of an immunosuppressive microenvironment in human metastatic PCa.

### 3.2. B Cell Proliferation and Activation Are Inhibited by a Low Dose of Ibrutinib through BTK Signaling

Through bioinformatics analysis and histological exploration, we have identified the critical role that BTK activity of TIL-Bs plays in the metastatic PCa microenvironment. Inhibition of B cell BTK activity probably reverses the resulting immunosuppressive microenvironment in the metastatic PCa microenvironment. Hence, we selected a commonly used BTK inhibitor, ibrutinib, and conducted an in vitro study to examine its inhibitory effect on BTK activity in B cells. Flow cytometry was used to evaluate the impact of ibrutinib on B cell proliferation and activation. As shown in Figure 2A, a low dose of ibrutinib (1 μM) significantly suppressed B cell proliferation under LPS stimulation. In addition, ibrutinib significantly decreased the numbers of CD19^+^CD80^+^ and CD19^+^IL-10^+^ cells (Figure 2B,C). Western blotting was conducted to detect the protein levels of B cell BTK signaling affected by ibrutinib. BTK phosphorylation is the first step in the BTK signaling cascade. A low dose of ibrutinib decreased the phosphorylation of BTK at Y223. The subsequent activations of the BTK signaling cascade by BTK phosphorylation, including phosphorylation of PLCγ2 at Y1217, ERK at T202/Y204, and NF-κB p65 at Ser536, were all decreased (Figure 2D, Appendix A). Thus, these results demonstrate the strong inhibitory effect of ibrutinib on B cell BTK signaling.

### 3.3. Ibrutinib Monotherapy Inhibits B Cell Infiltration and Activation in Mouse Orthotopic PCa Models, but Augments CD8^+^ T Cells with High Expression of Inhibitory Receptors (IRs) and Fails to Control Tumor Progression

In light of the above results, it is warranted to conduct in vivo study to verify the therapeutic effect of ibrutinib for metastatic PCa immunotherapy. We established a mouse orthotopic PCa model using the metastatic and hormone-independent PCa cell line—RM-1. The experimental scheme is illustrated in Figure 3A. Before starting the formal in vivo study, the optimum ibrutinib dose was explored and determined (6 mg/kg/d). The gross graph shows that the representative tumors in each group did not display a distinct change in size resulting from ibrutinib monotherapy (Figure 3B). Representative H&E and IHC images in each group exhibited similar cell morphology and Ki67 expression, indicating that tumor growth was not significantly affected by ibrutinib (Figure 3C,D). In addition, ibrutinib monotherapy did not considerably change tumor volume at the indicated time points, tumor weight at sacrifice, or tumor-bearing mouse survival (Figure 3E–G). Nevertheless, both total TIL-Bs and the activated B cell subset, as well as CD19^+^IL-10^+^ B cells, were significantly decreased in the ibrutinib group compared with the vehicle group (Figure 3H).

These results suggested that ibrutinib indeed suppressed B cell infiltration and function in mouse metastatic PCa models but failed to control tumor growth. We further explored the functional status of tumor killer cells—CD8^+^ T cells. As shown in Figure 4A, ibrutinib significantly elevated the proportion of tumor-infiltrating CD8^+^ T cells. In addition, these cells expressed higher PD-1 and CTLA4 but did not produce substantial IFN-γ. To further evaluate the effect of ibrutinib on BTK phosphorylation in TIL-Bs, we performed mIHC and confirmed the inhibition of BTK phosphorylation in the TIL-Bs of ibrutinib-treated tumor samples (Figure 4B). In concordance with the flow cytometry results, CD8^+^ T cell infiltration also increased in tumor samples from the ibrutinib-treated group (Figure 4C). Taken together, although it failed to retard tumor growth, ibrutinib monotherapy strikingly inhibited B cell infiltration and function and led to an increase in IRs highly expressing CD8^+^ T cells in the mouse PCa microenvironment.

### 3.4. The Low Dose of Ibrutinib Inhibits PCa Cell Viability but Not Apoptosis In Vitro and In Vivo

To evaluate the effect of a similar dose of ibrutinib on the proliferation of PCa cells, we performed a CCK8 assay to assess cell viability at the indicated time points. Human and mouse metastatic cell lines were treated with 0, 1, and 5 μM ibrutinib for 24, 48, and 72 h, respectively. As shown in Figure 5A, cell proliferation was significantly inhibited by 5 μM ibrutinib compared with the vehicle at each time point in both PC-3 and RM-1 cells. Although no statistical difference was found when the cells were treated with 1 μM ibrutinib at 24 and 72 h, distinguishable inhibition was also observed. Hence, these results suggest that a low dose of ibrutinib also exerts an inhibitory effect on PCa cell proliferation. To clarify whether the PCa cell reduction resulted from ibrutinib-induced apoptosis, we examined cell apoptosis by Annexin V/PI flow cytometry in vitro and surprisingly found that ibrutinib did not induce apoptosis of PCa cells per se at the same dose (Figure 5B). In addition, TUNEL staining was performed to evaluate apoptosis in the mouse tumor slides. Representative images of apoptotic cells are shown in Figure 5C; ibrutinib did not increase the number of apoptotic cells in the metastatic tumors. Collectively, these data suggested that a low dose of ibrutinib could inhibit PCa cell proliferation but was unable to induce cell apoptosis. Considering that B cells were more sensitive to ibrutinib in vitro, we continued to explore the changes in CD8^+^ T cell function induced by ibrutinib-treated B cells.

### 3.5. In Vitro Inhibition of BTK Signaling in B Cells by a Low Dose of Ibrutinib Promotes CD8^+^ T Cell Proliferation with Elevated PD-1, CTLA-4, and IFN-γ Expression

To assess CD8^+^ T cell proliferation influenced by ibrutinib-treated B cells, we designed an in vitro co-culture system and detected cell proliferation by CFSE staining using flow cytometry. Before B–T cells were co-cultured, purified B cells were treated with a low dose of ibrutinib (1 μM) for 72 h (Figure 6A). Seventy-two hours after B–T cells were co-cultured, CD8^+^ T cell proliferation was determined by detecting CD8^+^CFSE^+^ cells. As shown in Figure 6B, ibrutinib-treated B cells increased the proportion of CD8^+^CFSE^+^ cells to a great extent compared with the vehicle, indicating that CD8^+^ T cell proliferation was dramatically enhanced. Meanwhile, the expression of IRs, including PD-1 and CTLA4, was significantly elevated in CD8^+^ T cells, suggesting that an early exhausted phenotype was induced by ibrutinib-treated B cells. Interestingly, ibrutinib-treated B cells unexpectedly showed increased intracellular IFN-γ production in CD8^+^ T cells (Figure 6C). Above all, these results showed that ibrutinib-treated B cells promoted CD8^+^ T cell proliferation and probably switched them towards an early exhausted phenotype. Therefore, we conducted an in vivo study to investigate the therapeutic outcomes of combined ibrutinib and ICI treatment.

### 3.6. Ibrutinib Synergized with anti-PD-1 Therapy Strongly Enhances Antitumor Immunity in Murine Orthotopic PCa Models

Based on the increased infiltration of high-IRs CD8+ T cells caused by ibrutinib treatment in vivo and in vitro, we speculated that combined treatment with ibrutinib and ICIs would efficiently benefit monotherapy. To verify this hypothesis, we chose an anti-PD-1 monoclonal antibody (αPD-1) synergizing with ibrutinib to conduct an in vivo study. The experimental scheme is demonstrated in Figure 7A. As shown in Figure 7B,D,E, the tumor burden was significantly lower in the combined therapy group. H&E staining of tumor sections showed that the combined therapy led to a large scale of tumor destruction (Figure 7C). Survival analysis also showed that mice that received combined therapy had longer survival times than those receiving monotherapy or vehicle (Figure 7F). Flow cytometry plots showed that the total number and intracellular IFN-γ production of CD8^+^ T cells were both significantly enhanced by the combined therapy (Figure 7G). Representative mIHC images further confirmed the simultaneous suppression of BTK phosphorylation in TIL-Bs and augmentation of CD8^+^ T cell infiltration in tumors from mice who received the combined therapy when compared with those received αPD-1 monotherapy or vehicle (Figure 7H and Appendix A). In addition, we evaluated the side effects of ibrutinib administration on multiple organs by H&E staining. Compared with vehicle, ibrutinib did not cause certain pathological insults to vital organs, including the heart, liver, spleen, lungs, and kidneys (Appendix A). Overall, these results demonstrate that ibrutinib and anti-PD-1 monoclonal antibodies cooperated to strongly enhance antitumor immunity in a murine metastatic PCa model.

## 4. Discussion

Metastatic PCa is still considered to be incurable. Despite noticeable progress in systemic therapy, tumor heterogeneity and acquired resistance remain major obstacles restraining efficient remission and cure of metastatic PCa. Therefore, we chose metastatic and hormone-independent PCa cell lines (RM-1 and PC-3) for this study. Orthotopic implantation of RM-1 cells into the prostate of syngeneic mice results in an aggressive model of prostate cancer and metastatic activity in over 80% of animals by 16–17 days was documented, with the highest activity in the pelvic and retroperitoneal lymph nodes and the lowest in the lung [24]. Consistent with previous reports, we found the implanted prostate tumor invaded the surrounding tissues and bloody ascites in mice of the vehicle group, suggesting tumor metastasis. Although the animal model used in our study may not fully recapitulate the metastatic cascade observed in human prostate cancer, it is useful in testing immunotherapy for controlling late-stage prostate cancer. ICIs targeting T cell PD-1/PD-L1 axes for the treatment of various cancers have made scientific breakthroughs, but this approach has limited activity as monotherapy for late-stage PCa. Such a dilemma boosts investigations to explore combined strategies that synergize PD-1 checkpoint blockade with other agents for metastatic PCa therapy. As B cells have become a fascinating focus in immune tolerance, efforts have been made to seek reliable approaches to reverse B cell-acquired immunosuppressive properties inside the tumor immune microenvironment to improve immunotherapy. In this study, we discovered that an optimized dose of ibrutinib administration could ignite the “cold” status in the metastatic PCa microenvironment, creating a “hot” atmosphere for PD-1 blockade.

However, the overall functional role of TIL-Bs in cancer remains unclear. Previous studies have documented both protumor and antitumor properties exerted in the TME by TIL-Bs [25,26]. Regarding PCa, several TIL-B subsets have been identified as negative contributors to tumor regression [13,14,15,16]. In a recent clinical trial, direct depletion of total B cells by anti-CD20 monoclonal antibody treatment significantly decreased B cell density. Still, it simultaneously resulted in CD3 T cell reduction, which would probably undermine PD-1 checkpoint blockade efficacy [27]. Based on this result, we considered that anti-CD20 monoclonal antibody administration for TIL-B inhibition was unsuitable for the following combined immunotherapy. The BTK inhibitor ibrutinib was initially used to treat hematological and B cell malignancies. Now, it has been studied to treat solid malignancies [28]. Regarding B cells, ibrutinib inhibits BTK activation and disrupts B cell receptor (BCR) downstream signal transduction, suppressing B cell proliferation and differentiation [29]. This is consistent with our findings that a low dose of ibrutinib significantly inhibits B cell activation and proliferation in vitro. In the in vivo study, the optimized dose of ibrutinib also significantly reduced TIL-B density and inhibited function. It is rather remarkable that ibrutinib significantly reduced the production of IL-10 in B cells, a well-recognized regulatory B cell biomarker, which also suggested an alleviation of immunosuppressive effects brought by B cells [30,31]. In addition, ibrutinib can mechanically reverse Th2 cell polarization by inhibiting IL-2 inducible T cell kinase, which alters the potential for activation of Th1 and CD8^+^ T cells [32,33]. In our study, ibrutinib-treated B cells promoted CD8^+^ T cell proliferation in vitro. Moreover, ibrutinib increases CD8^+^ T cell infiltration in animal models. However, these CD8^+^ T cells showed high expression of IRs, suggesting a different mechanism of immune modulation of ibrutinib for B–T interaction. Regarding PCa cells per se, previous studies have shown that BTK expression is elevated in PCa cell lines and tumors. Consistent with our results, ibrutinib significantly inhibited the proliferation of PCa cell lines even at a low dose (1 μM). In addition, the low dose of ibrutinib failed to induce cancer cell apoptosis unless the concentration reached 30 μM [21,34]. As reported in the literature, the proper dose and route of ibrutinib administration in mice for immunotherapeutic studies are injected intraperitoneally and 6 mg/kg once per day [35,36]. We have tried higher doses of ibrutinib in murine models to achieve dual effects—inhibition of both B cells and cancer cells—but unexpectedly accelerated animal death. Therefore, we followed the dose of ibrutinib used in previous studies in this study. Upon treatment, the PCa tissues did not exhibit significant apoptosis induced by ibrutinib monotherapy. Surprisingly, the expression of IRs such as PD-1 and CTLA-4 in tumor-infiltrated CD8^+^ T cells was simultaneously increased, which would facilitate ICI application. Bearing in mind that tumor-infiltrated CD8^+^ T cells were expanded with high expression of PD-1, we combined ibrutinib with PD-1 checkpoint blockade and successfully achieved apparent tumor regression in animal models.

Another vital rationale behind the use of ibrutinib in solid tumors worthy of mention is that ibrutinib is not entirely selective towards BTK. It also exerts action by irreversibly inhibiting other kinases, such as epidermal growth factor receptor (EGFR), human epidermal growth factor receptor 2 (ERBB2/HER2), and Janus kinase 3 (JAK3), which have been proven to promote the development of solid tumors [28]. In terms of limitations, using an animal model may not comprehensively elucidate the mechanisms of ibrutinib efficacy in metastatic PCa immunotherapy. Subsequent investigations, including laboratory and clinical experiments, are needed to verify these findings further before they can be applied to human treatment.

## 5. Conclusions

Our preclinical data demonstrated that ibrutinib potentiates anti-PD-1 blockade therapeutic efficacy by inhibiting B cell infiltration and immunosuppressive effects through the BTK pathway (Figure 8). These findings provide important insights into the use of kinase agents in combination with ICIs to treat metastatic PCa. Our study will be helpful for the development of novel strategies for metastatic or late-stage PCa immunotherapy.

## Figures and Tables

**Figure 1 cancers-15-02356-f001:**
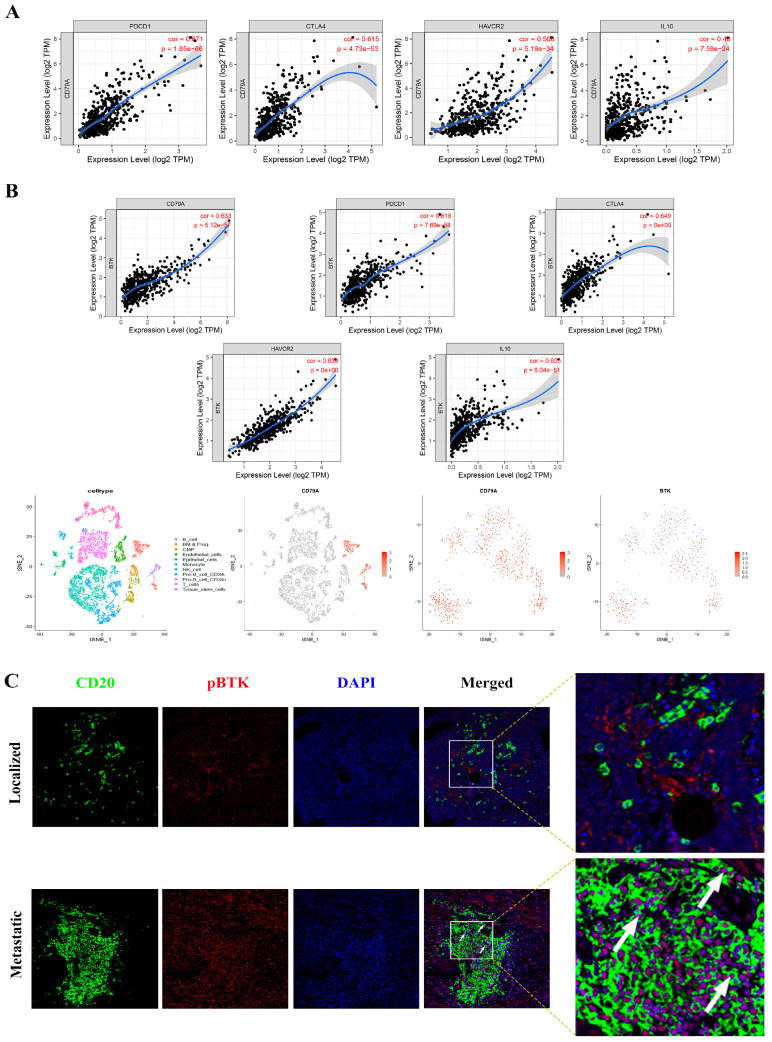
Bruton’s tyrosine kinase (BTK) is largely expressed in tumor−infiltrating B cells and associated with immunosuppressive molecules in human metastatic prostate cancer (PCa). (**A**) Correlation analysis of gene expression in PCa tissues from a public database (TIMER) reveals that CD79A (commonly known as a B cell marker) is positively associated with the expression of PDCD1, CTLA4, HAVCR2, and IL-10. (**B**) BTK expression is positively associated with the expression of CD79A, IL-10, and PDCD1. Additionally, single-cell RNA-seq analysis from a published dataset (GSE143791) containing metastatic PCa samples revealed that BTK is highly expressed in B cells (CD79A) infiltrated in metastatic PCa. (**C**) Representative mIHC images show that the phosphorylation of BTK in tumor-infiltrating B cells is mainly observed in metastatic PCa. The white arrows indicate the BTK-phosphorylated B cells (original magnification: 400×). A *p* value less than 0.05 is statistically significant.

**Figure 2 cancers-15-02356-f002:**
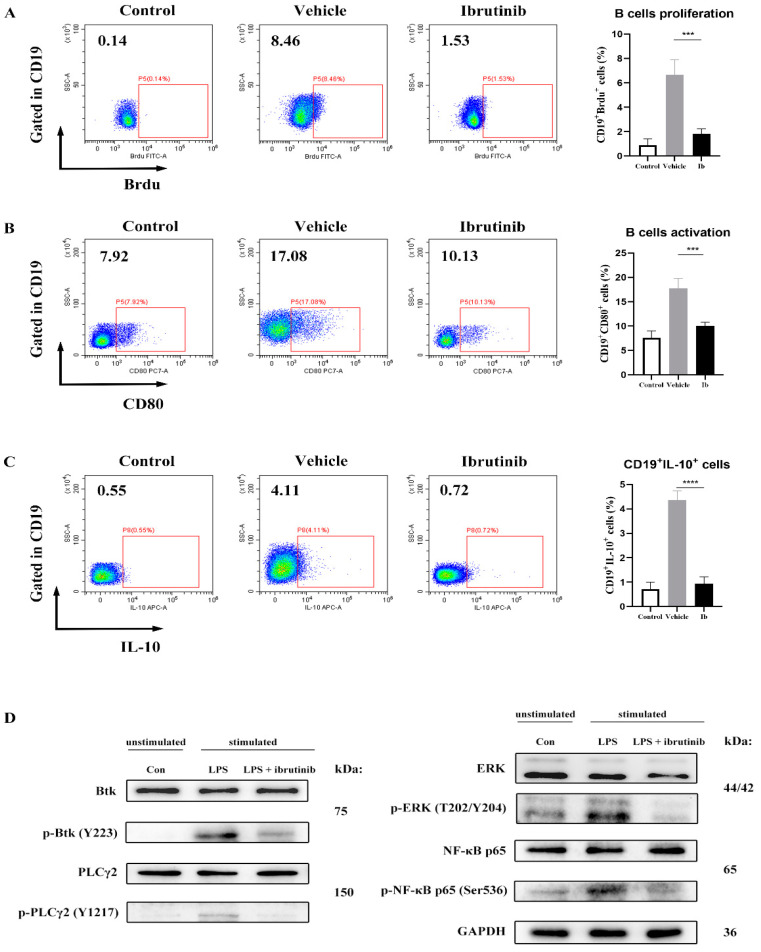
In vitro B cell activation and proliferation are suppressed by a low dose of ibrutinib through inhibiting BTK signaling. Mouse splenic B cells were purified and isolated using antimouse CD19 magnetic microbeads and the AutoMACS program. Purified B cells were stimulated with 10 μg/mL LPS and treated with 1 μM ibrutinib or DMSO for 72 h. (**A**–**C**) B cell proliferation, activation, and IL-10 production were determined by flow cytometry. (**D**). Western blot assays were performed to analyze the protein levels of the BTK signaling pathway in B cells. A *p* value less than 0.05 was considered statistically significant. ***, *p* < 0.001; ****, *p* < 0.0001. The uncropped bolts are shown in Appendix A.

**Figure 3 cancers-15-02356-f003:**
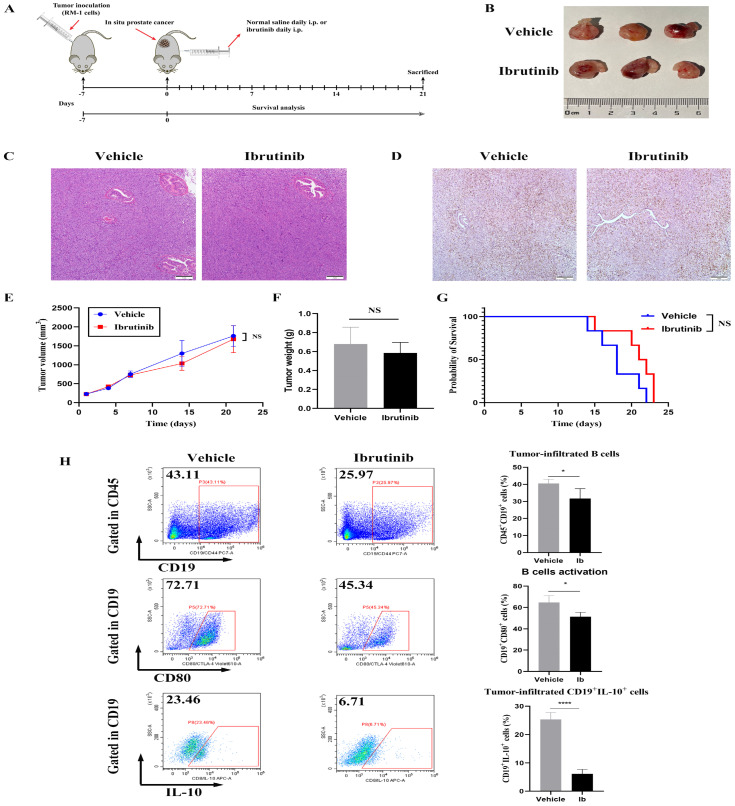
In vivo ibrutinib monotherapy does not suppress PCa growth but hinders TIL-B infiltration and activation. (**A**) Experimental design and treatment strategy. See Materials and Methods for details. (**B**) Representative photographs displaying the gross appearance of RM-1 tumors in mice when sacrificed. (**C**) H&E staining shows inconspicuous pathological changes after treatment with ibrutinib (original magnification: 100×). (**D**) IHC staining shows unchanged Ki67 expression in both groups (original magnification: 100×). (**E**) Tumor volumes of mice in both groups were recorded at the indicated time points (*n* = 4). (**F**) Tumor weights of mice in both groups were measured when sacrificed (*n* = 4). (**G**) Kaplan-Meier survival curve of tumor−bearing mice treated as indicated (*n* = 6). (**H**) Flow cytometry analysis of TIL-Bs. Ibrutinib administration significantly inhibited the infiltration and activation of TIL-Bs in mouse orthotopic PCa models (*n* = 4). A *p* value less than 0.05 was considered statistically significant. TIL-Bs, tumor−infiltrating B cells. *, *p* < 0.05; ****, *p* < 0.0001.

**Figure 4 cancers-15-02356-f004:**
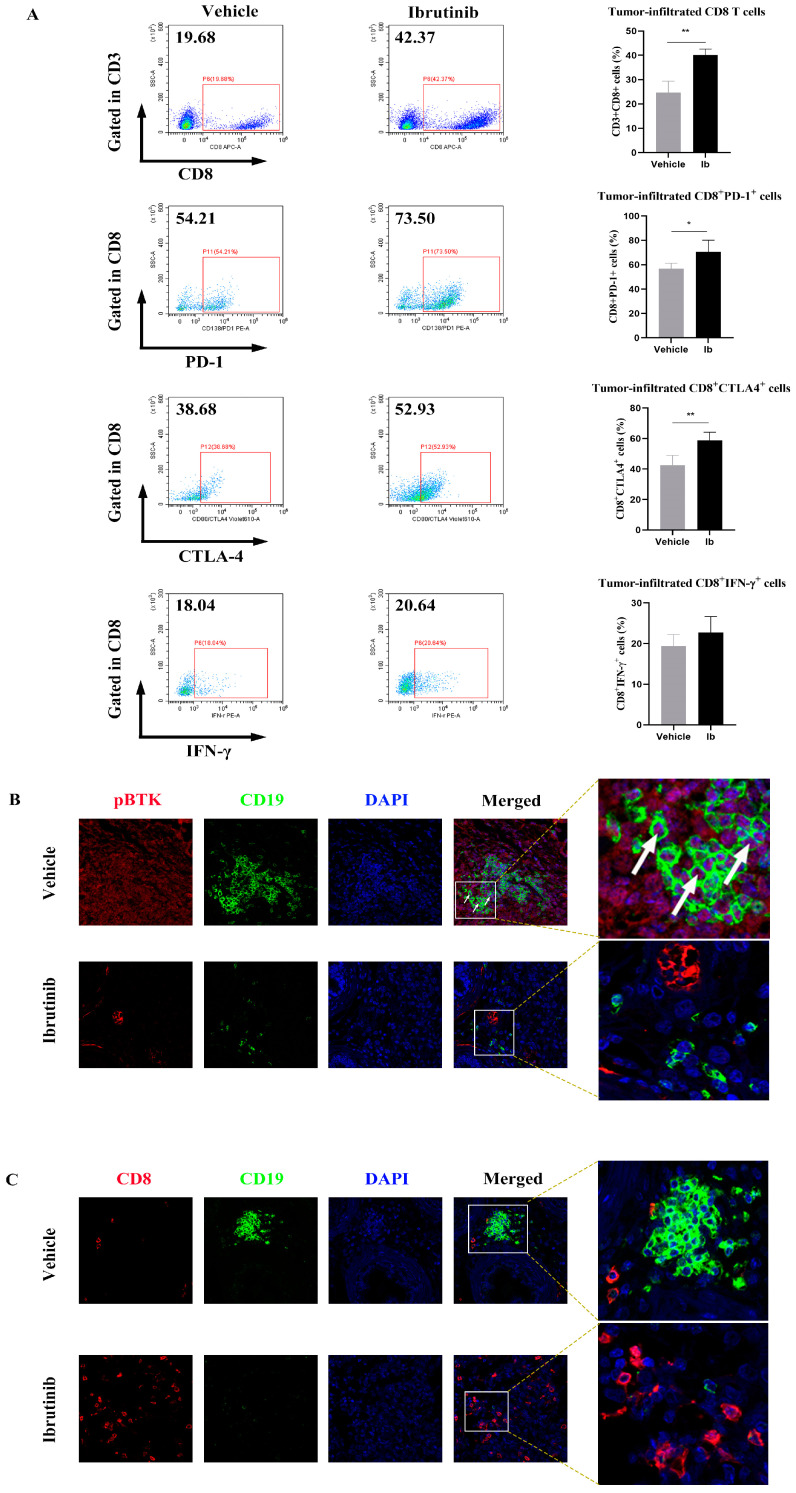
In vivo ibrutinib monotherapy leads to an increase in high-IRs CD8^+^ T cell infiltration in PCa. (**A**) Flow cytometry analysis of tumor−infiltrated CD8^+^ T cells in PCa. The plots show that CD8^+^ T cells were increased with high expression of PD-1 and CTLA4 but IFN-γ production was unchanged by ibrutinib monotherapy (*n* = 4). (**B**,**C**) Representative mIHC images of mouse PCa tissues showing changes in CD19^+^ B cell (green) and CD8^+^ T cell (red) infiltration after treatment (original magnification: 400×). A *p* value less than 0.05 was considered statistically significant. IRs, inhibitory receptors. *, *p* < 0.05; **, *p* < 0.01.

**Figure 5 cancers-15-02356-f005:**
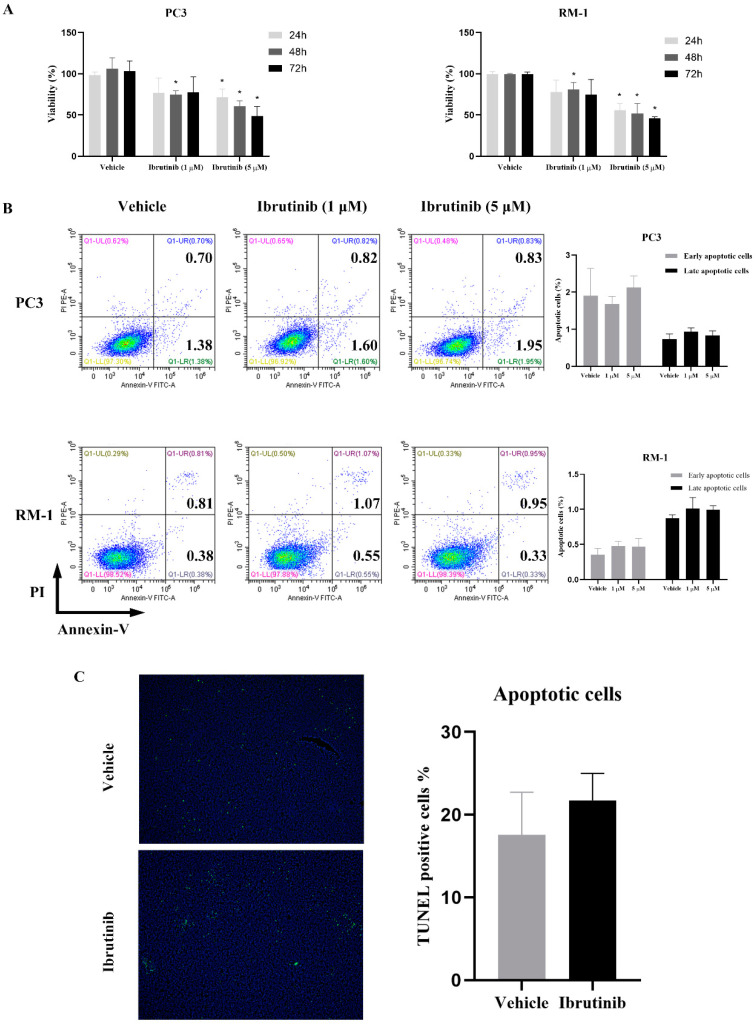
Ibrutinib inhibits PCa cell viability in vitro but does not induce comparable apoptosis at the indicated doses. (**A**) Human (PC-3) and mouse (RM-1) PCa cells were seeded in 96-well plates and treated with the specified concentrations (0, 1, and 5 μM) of ibrutinib for 24, 48, and 72 h. A CCK8 assay was used to evaluate cell viability. (**B**) PC-3 and RM-1 cells were seeded in 24-well plates and treated with ibrutinib at 0, 1, and 5 μM for 48 h. The cells were stained with PI (PE) and Annexin V (FITC), followed by flow cytometry detection. (**C**) Representative images of TUNEL staining and the percentage of TUNEL−positive cells (green) in mouse PCa tissues after treatment (*n* = 4, original magnification: 100×). A *p* value less than 0.05 was considered statistically significant. *, *p* < 0.05.

**Figure 6 cancers-15-02356-f006:**
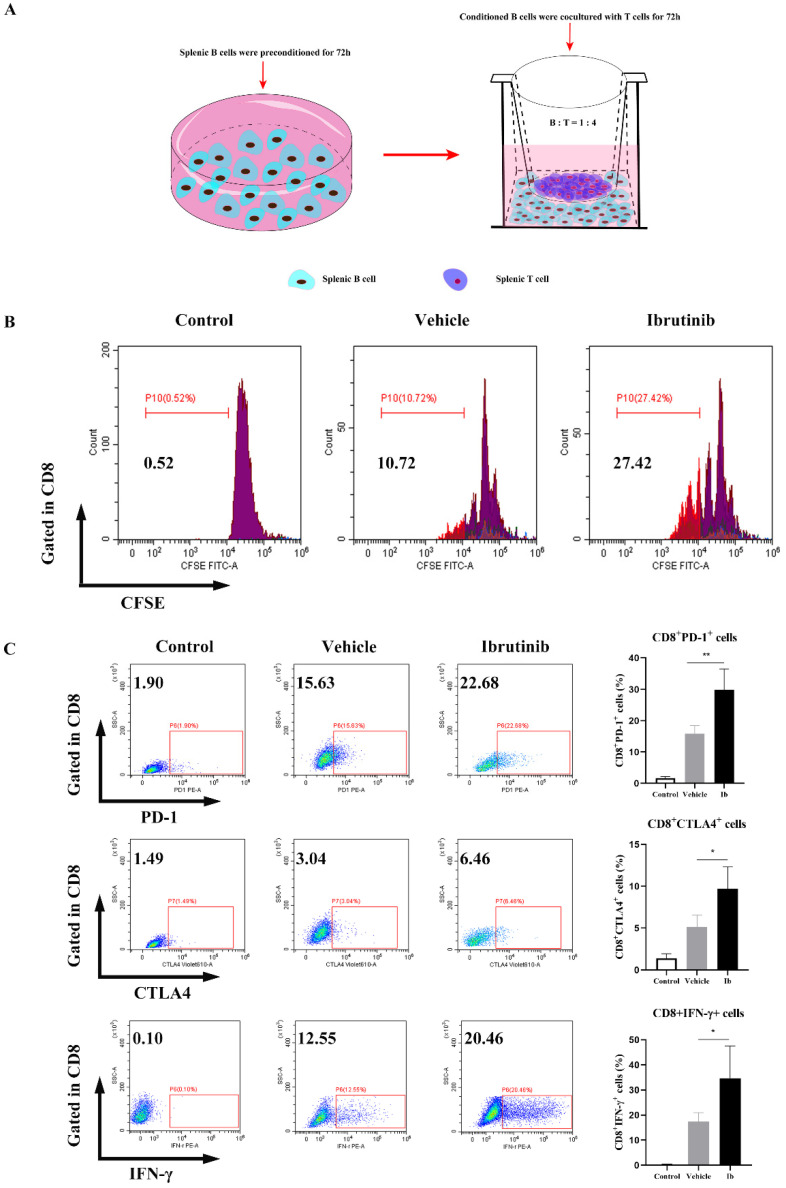
Ibrutinib-treated B cells promote the proliferation of CD8^+^ T cells with elevated PD-1, CTLA4, and IFN-γ expression in vitro. (**A**) Illustration of the culture system. The conditioned B cells were co-cultured with T cells stimulated with anti-CD3 and anti-CD28 antibodies for 72 h. (**B**) Representative flow cytometry plots showing CD8^+^ T cell proliferation by CFSE staining after co-culturing with conditioned B cells for 72 h. (**C**) Representative flow cytometry plots of PD-1, CTLA4, and IFN-γ expression in CD8^+^ T cells after co−culturing with conditioned B cells for 72 h. A *p* value less than 0.05 was considered statistically significant. *, *p* < 0.05; **, *p* < 0.01.

**Figure 7 cancers-15-02356-f007:**
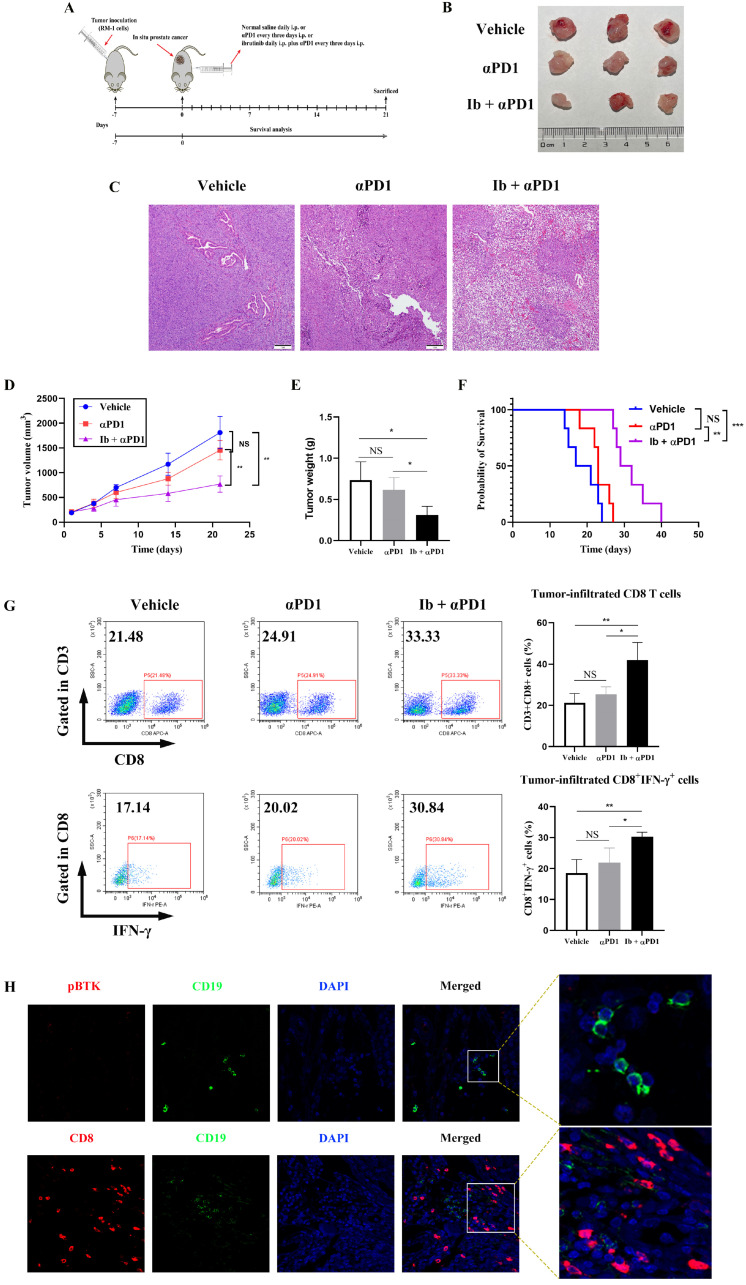
Ibrutinib synergizes with anti-PD-1 therapy to achieve great tumor control. (**A**) Experimental design and treatment strategy. See Materials and Methods for details. (**B**) Representative photographs show the gross appearance of RM-1 tumors in mice in each group. (**C**) HE staining shows noticeable pathological changes after combined treatment (original magnification: 100×). (**D**,**E**) Tumor volumes at the indicated times and weights were recorded in the three groups (*n* = 4). (**F**) Survival of tumor-bearing mice in each group (*n* = 6). (**G**) Flow cytometry plots show that the combined treatment enhanced the augmentation of tumor-infiltrating CD8^+^ T cells and CD8+IFN-γ^+^ cells (*n* = 4). (**H**) Representative mIHC images display inhibition of BTK phosphorylation in B cells and augmentation of CD8^+^ T cells in tumors of mice treated with ibrutinib plus αPD-1 (original magnification: 400×). A *p* value less than 0.05 was considered statistically significant. *, *p* < 0.05; **, *p* < 0.01; ***, *p* < 0.001.

**Figure 8 cancers-15-02356-f008:**
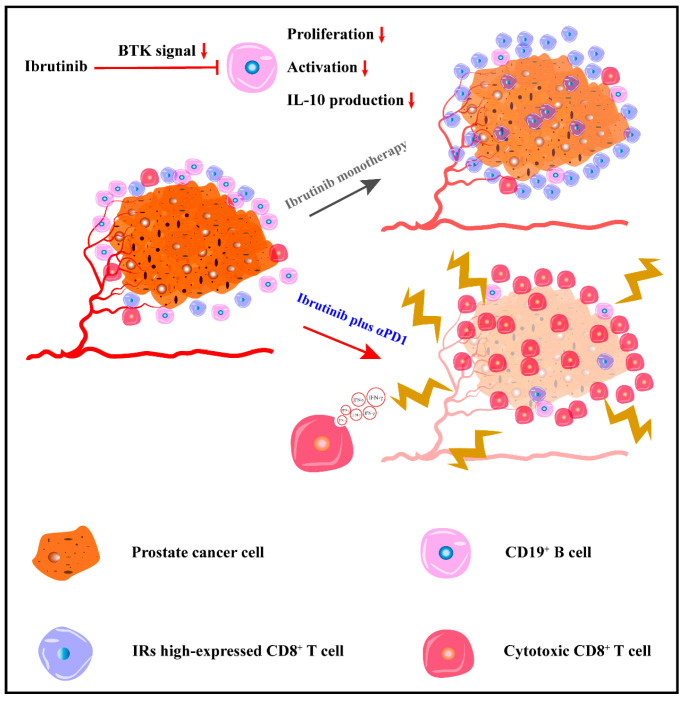
Conceptual graph. Ibrutinib monotherapy fails to suppress mouse metastatic PCa growth but inhibits B cell proliferation, activation, and IL-10 production, leading to the augmentation of CD8^+^ T cells highly expressing IRs through the BTK signaling. Ibrutinib synergizes with anti-PD-1 immunotherapy, elicits an enhanced cytotoxic response of CD8^+^ T cells, and achieves great control of mouse metastatic PCa. BTK, Bruton’s tyrosine kinase; IRs, inhibitory receptors; PCa, prostate cancer.

## Data Availability

The public data included in this study are available in online databases, including TIMER (https://cistrome.shinyapps.io/timer/, accessed on 25 January 2022) and TISCH (http://tisch.comp-genomics.org/, accessed on 25 January 2022).

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
