# Peer review of "Ibrutinib Inhibits BTK Signaling in Tumor-Infiltrated B Cells and Amplifies Antitumor Immunity by PD-1 Checkpoint Blockade for Metastatic Prostate Cancer"

_cancers, 2023, doi:10.3390/cancers15082356_

Round 1

Reviewer 1 Report

Cancers-2259145

Ibrutinib Inhibits BTK Signaling in Tumor-infiltrated B cells and Amplifies Antitumor Immunity by PD-1 Checkpoint 3 Blockade for Metastatic Prostate Cancer

Summary: In this study the authors provide a novel combinational regimen with a kinase inhibitor and PD-1/PD-L1 axis blockade in metastatic prostate cancer. The authors have shown that Ibrutinib increases the effect of immunotherapy and decreases tumor growth in a murine metastatic prostate cancer model.

Comments:

1. Fig. 1A: The authors performed the gene correlation analysis and observed that CD79A expression was significantly associated with PDCD1, CTLA4, HAVCR2, and IL10, which are commonly recognized as immunosuppressive molecules. Are they related to the BTK pathway? The authors should explain this in the text. 

2. Fig. 1C: The authors claim that pBTK is mainly observed in metastatic PCa tissues compared to localized samples; but only CD20 which is a B cell marker showed a significant difference. pBTK staining in both samples is not clear enough to reach any conclusion. The authors should provide better quality pictures and additional experimental proof to support this claim.

3. Fig. 3: According to this data, a low dose of Ibrutinib treatment had no effect on primary tumor growth. The authors can show some cell proliferation marker and pBTK, pPLCγ expression in tumor sections to support their hypothesis.

4. Fig. 4B: Again, the mIHC data showing pBTK is not very convincing, the authors should provide better quality pictures.

5. In this study, the authors used an orthotopic PC model using RM1 cells but there is insufficient evidence for multi-organ metastasis.  The authors should include the metastatic PC model to explore the effect of Ibrutinib and anti-PD1 in different organ metastatic tumors. As the authors mention, in metastatic PCa, pBTK expression is greater as compared to primary tumors. The authors should check the effect of Ibrutinib in different metastatic organ sites.

Author Response

Dear Reviewer,

Thank you for taking the time to review our submission. We appreciate your valuable feedback and comments.

We have carefully considered your suggestions and made the necessary revisions to improve the quality of our work. Please find attached our response to your comments, which outlines the changes we have made to our manuscript. We believe that these changes have strengthened our paper and addressed the concerns that you raised.

We hope that you find our revised manuscript satisfactory and that you can recommend it for publication. Once again, we are grateful for your constructive feedback, which has helped us to improve our work.

Thank you for your time and consideration.

Sincerely,

Jinming Di, M.D., Ph.D.

The Third Affiliated Hospital of Sun Yat-sen University, Guangzhou, China

600 road Tianhe, Tianhe District, Guangzhou 510630, China

Email address: [email protected]

Reviewer 2 Report

Line 102: The word “effects” looks redundant here.

Line 102: The last sentence of the introduction belongs to the discussion or conclusion.

Line 107 to 114. Informed consent was mentioned two times.

Line 153: A space is missing before penicillin.

Line 218: CO2 must be written like the rest of the manuscript.

Line 274 and 311 and 336 and 363 and 411 P value, the p shouldn’t be capitalized here.

Please have consistency in choosing between minutes or mins throughout the text.

Figure 1, A and B are not readable and have terribly low quality in this size.

Overall, the excessive use of abbreviations made reading the manuscript, especially the introduction, difficult. For example, ADT in line 50 was not used again, and some abbreviations like IFN-gamma were not introduced. I recommend addressing these issues. 

Reviewer 3 Report

Although the current study is interesting, there is a serious shortcoming:

Most parts of the initial results are based on in silico studies. However, the authors should confirm their findings based on the wet-lab studies. For example, the authors should isolate tumor-infiltrated B cells and treat them with Ibrutinib in vitro and assess cytokine production and inhibitory molecules expression pattern. 

Author Response

Dear Reviewer,

Thank you for reviewing our submission and providing us with insightful feedback.

We have taken your comments into consideration and made the necessary revisions to address the issues you raised. Please find attached our response to your comments, which outlines the changes we have made to our manuscript.

We would like to express our gratitude for your time and effort in reviewing our work. Your feedback has been immensely helpful in improving the quality of our manuscript. We hope that the revisions we have made will meet your expectations and that you will find our revised manuscript suitable for publication.

Thank you again for your valuable feedback and support.

Best regards, 

Jinming Di, M.D., Ph.D.

The Third Affiliated Hospital of Sun Yat-sen University, Guangzhou, China

600 road Tianhe, Tianhe District, Guangzhou 510630, China

Email address: [email protected]

Round 2

Reviewer 1 Report

Cancers-2259145R

Ibrutinib Inhibits BTK Signaling in Tumor-infiltrated B cells and Amplifies Antitumor Immunity by PD-1 Checkpoint 3 Blockade for Metastatic Prostate Cancer

Summary: The authors attempted to address the previous comments; however the following few comments need further clarification.

Comments:

1. Fig. 1A: The authors performed the gene correlation analysis and observed that CD79A expression was significantly associated with PDCD1, CTLA4, HAVCR2, and IL10, which are commonly recognized as immunosuppressive molecules. Are they related to the BTK pathway? The authors should explain this in the text. 

The authors added some discussion in the response letter regarding this point which should be added to the paper.

2. Fig. 1C: The authors claim that pBTK is mainly observed in metastatic PCa tissues compared to localized samples; but only CD20 which is a B cell marker showed a significant difference. pBTK staining in both samples is not clear enough to reach any conclusion. The authors should provide better quality pictures and additional experimental proof to support this claim.

Authors provided new mIHC data for pBTK (red) but still it is not very convincing since the red staining is very low in metastatic sample. Please try to provide more convincing images. Figure S3A actually looks more convincing and can be added to the main figures. A similar comment implies to Fig. 4B. The red staining (pBTK) is too weak in vehicle treated samples.     

3. In this study, the authors used an orthotopic PC model using RM1 cells but there is insufficient evidence for multi-organ metastasis. The authors should include the metastatic PC model to explore the effect of Ibrutinib and anti-PD1 in different organ metastatic tumors. As the authors mention, in metastatic PCa, pBTK expression is greater as compared to primary tumors. The authors should check the effect of Ibrutinib in different metastatic organ sites.

The authors added some arguments in support of their claim of using this model as a metastatic model in their response letter, which should be added in the paper. Also, the authors state “we found the implanted prostate tumor invaded the surrounding tissues and bloody ascites in our animal models, suggesting tumor metastasis”; do the authors see any decrease in the metastasis/invasion to the surrounding tissue and ascites contents in Ibrutinib and anti-PD1 treated groups as compared to the vehicle treated group? Also was the pBTK expression different between localized PCa vs. the surrounding tissue?

4. Figure 7H:  are both the upper and lower panel from tumors of mice treated with ibrutinib plus αPD-1? How about monotherapy or vehicle treatment group? How the staining patterns look in those tumor samples?

Reviewer 3 Report

My comment was properly addressed. 

Author Response

Thank you for taking the time to review our manuscript. I appreciate the feedback you provided and I am pleased to hear that you feel your comments have been properly addressed.